# Effects of an Exercise Program Combining Aerobic and Resistance Training on Protein Expressions of Neurotrophic Factors in Obese Rats Injected with Beta-Amyloid

**DOI:** 10.3390/ijerph19137921

**Published:** 2022-06-28

**Authors:** Gyuho Lee, Yunwook Kim, Jung-Hee Jang, Chan Lee, Jaewoo Yoon, Nayoung Ahn, Kijin Kim

**Affiliations:** 1Department of Physical Education, College of Physical Education, Keimyung University, Daegu 42601, Korea; judo7521@hanmail.net (G.L.); a2461078@gmail.com (Y.K.); nyahn13@kmu.ac.kr (N.A.); 2Department of Pharmacology, School of Medicine, Keimyung University, Daegu 42601, Korea; pamy202@kmu.ac.kr (J.-H.J.); leechan777@naver.com (C.L.); 3College of Pharmacy, Keimyung University, Daegu 42601, Korea; jwyoon@kmu.ac.kr

**Keywords:** β-amyloid, obesity, exercise, BDNF, brain function

## Abstract

In this study, the effects of a 12-week exercise program combining aerobic and resistance training on high-fat diet-induced obese Sprague Dawley (SD) rats after the injection of beta-amyloid into the cerebral ventricle were investigated. Changes in physical fitness, cognitive function, blood levels of beta-amyloid and metabolic factors, and protein expressions of neurotrophic factors related to brain function such as BDNF (brain-derived neurotrophic factor) in the quadriceps femoris, hippocampus, and cerebral cortex were analyzed. The subjects were thirty-two 10-week-old SD rats (DBL Co., Ltd., Seoul, Korea). The rats were randomized into four groups: β-Non-Ex group (*n* = 8) with induced obesity and βA25-35 injection into the cerebral ventricle through stereotactic biopsy; β-Ex group (*n* = 8) with induced obesity, βA25-35 injection, and exercise; S-Non-Ex group (*n* = 8) with an injection of saline in lieu of βA25-35 as the control; and S-Ex group (*n* = 8) with saline injection and exercise. The 12-week exercise program combined aerobic training and resistance training. As for protein expressions of the factors related to brain function, the combined exercise program was shown to have a clear effect on activating the following factors: PGC-1α (peroxisome proliferator-activated receptor gamma coactivator 1-alpha), FNDC5 (fibronectin type III domain-containing protein 5), and BDNF in the quadriceps femoris; TrkB (Tropomyosin receptor kinase B), FNDC5, and BDNF in the hippocampus; PGC-1α, FNDC5, and BDNF in the cerebral cortex. The protein expression of β-amyloid in the cerebral cortex was significantly lower in the β-Ex group than in the β-Non-Ex group (*p* < 0.05). The 12-week intervention with the combined exercise program of aerobic and resistance training was shown to improve cardiopulmonary function, muscular endurance, and short-term memory. The results demonstrate a set of positive effects of the combined exercise program, which were presumed to have arisen mainly due to its alleviating effect on β-amyloid plaques, the main cause of reduced brain function, as well as the promotion of protein expressions of PGC-1α, FNDC5, and BDNF in the quadriceps femoris, hippocampus, and cerebral cortex.

## 1. Introduction

The physiological mechanisms underlying the development of neurodegenerative disease remain unclear [1], but the main causal factor is presumed to be beta-amyloid plaques, which increase the probabilities of synaptic dysfunction due to accelerated neuronal damage and of Alzheimer’s Disease as a degenerative phenomenon in the nervous system [2]. A decrease in the concentration of beta-amyloid was found to improve the synapse function and enhance cognitive function while delaying the onset of Alzheimer’s disease. Neurodegenerative diseases associated with beta-amyloid plaques show strong correlations with obesity, higher insulin resistance, metabolic diseases [3], and low androgen and estrogen levels [4,5]. Hence, efforts to reduce beta-amyloid concentrations in the elderly and aging population could represent an important target for the prevention and treatment of dementia. Regular exercise has a positive effect in terms of preventing cognitive dysfunction and degeneration of the brain cortex and hippocampus, as well as in delaying memory loss and enhancing antioxidant activity [6]. In addition, by reducing beta-amyloid plaques, regular exercise may contribute to the prevention [7] and improvement [8] of neurodegenerative diseases such as Alzheimer’s disease and Parkinson’s disease. Furthermore, exercise induces the expression of neurotrophic factors and activation of synaptic plasticity in the brain through changes such as increasing hippocampal or callosal size, activating blood flow, altering axons and dendrites, and activating neurogenesis [9]. The increased expressions of neurotrophic factors, in particular, which occur in the brain after exercise, cause a clear increase in the expression of the D2 dopamine receptor in the hippocampal dentate gyrus and cerebral cortex [9]. In a study on patients with ataxia, exercise was reported to have a protective effect on the brain as the serum levels of neurotrophic factors including BDNF were increased and the secretion of neurotransmitters was activated [6]. The results of such studies provide critical evidence for supporting the non-pharmacological effects of exercise in the prevention or treatment of neurodegenerative diseases and dementia. Nonetheless, the molecular and biological mechanisms in the nervous system underlying the protective effect of exercise on brain function have not yet been elucidated.

Aerobic exercise has been reported to activate mitochondrial biogenesis and angiogenesis [10], as well as protein expressions of factors that promote brain function, namely PGC-1α, AMPK (5′ adenosine monophosphate-activated protein kinase), irisin, and BDNF [11]. In a study on rodents, exercise on a treadmill at 60–70% intensity was shown to enhance the function of vascular endothelial cells and reduce levels of the inflammatory markers IL-6 (interleukin-6) and TNF-α (tumor necrosis factor-α) [12]. Aerobic exercise is also helpful for activating neuroglial cells in brain tissues, alleviating chronic inflammation [13], and enhancing the functioning of the central nervous system [14]. Resistance exercise is known to be helpful for the maintenance of muscle mass and strength as it activates myokine secretion [15], and also contributes to the prevention of cognitive dysfunction through the inhibition of lipid peroxidation and acetylcholine activation in the hippocampus [12]. Nevertheless, whether or not resistance exercise has positive effects in terms of reducing beta-amyloid plaques or activating factors that promote brain function is unclear. It is noteworthy that the combination of aerobic and resistance exercise has been shown to contribute to the prevention of sarcopenia, a key factor in the maintenance of health as one ages, and in the activation of cardiopulmonary function, while exerting parallel effects on the activation of the metabolic functions of factors related to brain function and on the secretion of myokines. In addition, it is required to combine high-intensity aerobic exercise and resistance exercise when performing exercise to prevent brain function deterioration in the aging process [16,17]. Recently, however, Liang et al. [18] pointed out that, despite the clear effects of exercise on enhancing cognitive function in patients with Alzheimer’s disease, evidence on the form, frequency, and intensity of exercise remains insufficient, and no in-depth analysis on this topic has been reported. Thus, a study on the effects of an exercise program combining aerobic and resistance training on the alleviation of beta-amyloid plaques, the central factor in the development of neurodegenerative diseases, and on the specific mechanisms of the protective effects on brain function may contribute to the elucidation of effective exercise programs for promoting brain function.

Therefore, in this study, we investigated changes in physical fitness, cognitive function, blood levels of beta-amyloid and metabolic factors, and protein expressions of neurotrophic factors related to brain function such as BDNF in the quadriceps femoris, hippocampus, and cerebral cortex, after injecting beta-amyloid into the cerebral ventricle of high-fat diet-induced obese SD rats and subjecting them to a 12-week exercise program combining aerobic and resistance training.

## 2. Materials and Methods

### 2.1. Subjects

The subjects in this study were thirty-two 10-week-old Sprague Dawley (SD) rats (DBL Co., Ltd., Seoul, Korea). All rats were allowed a period of adaptation to the environment through a week of preliminary breeding. The rats were then fed a high-fat diet for 6 weeks to induce obesity, followed by a stereotactic biopsy and postoperative injection of βA25-35 for two weeks. During the breeding period, the daily light-dark cycle was set to 12 h (from 7 a.m. to 7 p.m.) and the temperature and relative humidity in the breeding room were maintained at approximately 24 ± 1 °C and 60%, respectively. The rats were split into four treatment groups: β-Non-Ex group (*n* = 8) with induced obesity and βA25-35 injection; β-Ex group (*n* = 8) with induced obesity, βA25-35 injection, and exercise; S-Non-Ex group (*n* = 8) with injection of saline in lieu of βA25-35 as the control; and S-Ex group (*n* = 8) with saline injection and exercise. The study protocol was approved by the Institutional Animal Care and Use Committee at BioHealth Convergence Center, Daegu Technopark (Approval No. BHCC-IACUC-2017-01).

### 2.2. Experimental Procedures

In this study, a preliminary analysis of cognitive function was performed using the passive avoidance test to assess short-term memory and the water maze test to assess spatial learning ability in obesity-induced SD rats. After the preliminary tests, the rats underwent the stereotactic biopsy for cranial perforation in the area connecting to the cerebral ventricle, and according to their respective experimental groups, were injected with either βA25-35 or saline. The Ex groups were guided to follow a combined exercise program of aerobic and resistance training after the injection of βA25-35 or saline, and after 12 weeks, a post-intervention cognitive function test was performed. For the between-group comparison of physical fitness, muscular endurance and cardiopulmonary function were tested, and after a 12-h period of fasting, the change in blood glucose concentration was examined using the OGTT (oral glucose tolerance test). At the end of the experiment, the rats were sacrificed after a 12-h fasting period for sampling. The collected tissues and blood were immediately freeze-dried and stored at −70 °C. Figure 1 presents a diagram of the experimental procedures.

### 2.3. Dietary Composition

The composition of the high-fat diet (Doo Yeol Biotech. Co., Ltd., Seoul, Korea) was 30% carbohydrate, 50% fat, and 20% protein, with the addition of vitamins (19 g/kg Teklad Vitamins Mix No. 94047), minerals (43 g/kg Teklad Mineral Mix No. 94046) and choline bitartrate (3.0 g/kg). The rats were allowed liberal amounts of drinking water and feed.

### 2.4. Exercise Program

The exercise program began with resistance training from 9:00–10:00 and then proceeded to the aerobic training. For the resistance training, with the aim of increasing muscular strength and counteracting the effects of obesity and aging, the method developed by Lee & Kim [19] was used after being modified to suit the experimental conditions in this study. The rats performed a climbing exercise on a ladder with a weight attached to their tails. For aerobic training, the method described in Kim et al. [20] was used after partial modification. The rats ran for 30 min at 25 m/min on a motorized treadmill (Quinton Instrument, Seattle, WA, USA), with the slope fixed at 0°.

### 2.5. Stereotactic Biopsy for βA25-35 Injection

For βA25-35 injection, a stereotactic biopsy was performed after anesthetizing the rat with Zoletil (0.04–0.06 mL/kg) and Rompun (0.12 mL/kg). The head of the rat was shaved and fixed horizontally on an animal stereotactic frame (Leica Co., Ltd., Weitzlar, Germany) for stereotactic surgery. The scalp was incised and the fascia and foreign substances were removed, the coordinates were set, and the perforation was generated. After a day of rest, the rat was injected with 5 μL of βA25-35 or saline at a rate of 1 μL/min, into the V–8.0 mm area from the endocranial membrane using an automated micro syringe pump (Harvard Apparatus, Holliston, MA, USA) with a 25 μL Hamilton syringe (Hamilton company, Reno, NV, USA) combined with a 33-gauge microneedle. For βA25-35, the Beta Protein 25–35 supplied by BACHEM Co., Ltd. was used. The method described by Ghasemi, Zarifkar, Rastegar, Maghosudi & Moosavi [21] was used to produce the culture to induce the coagulation of βA25-35: four days in a water bath at 37 °C with subsequent addition of distilled water to produce a 1 mM stock to be stored at −20 °C.

### 2.6. Experimental Variables and Methods

Cognitive function test: In the cognitive function test, short-term memory and spatial learning ability were measured. The passive avoidance test was performed to assess short-term memory. For this test, the rat was placed in a transparent box called the Shuttle Box (Jeungdo Bio & Plant Co., Ltd., Seoul, Korea). After a period of adaptation, the light was switched on and after approximately one minute, the partition was removed to reveal a free passageway. Then, the time between the removal of this partition and the complete migration of the rat (including all of its four feet) from the transparent box to the dark box was measured [22]. For spatial learning ability, the water maze test by Morris [23] was performed. After a period of adaptation to the platform inside a water tank, the time taken for the rat to spot and reach a target object was measured.

OGTT: After 48 h of rest and 12 h of fasting, blood was collected from the tail of the rat at rest, and after oral administration of a 50% glucose solution, 400 μL of blood was collected from the rat at 15, 30, 60, and 120 min. After the anticoagulation treatment of the collected blood (with 50 μL of heparin), the blood was centrifuged (1500× *g* for 15 min), and blood glucose concentration was measured from the serum.

Physical fitness test: In the muscular endurance test, the rat was made to hang on a bar (5 cm width) at 50 cm height above the ground, and the holding time (sec) was measured. For the cardiopulmonary function test, the rat was made to run at 32 m/min on a treadmill with an 8° slope, and the time until complete exhaustion was measured.

Animal sacrifice and sampling: At 48 h after completion of the exercise, the rat was anesthetized with Zoletil 50 (10 mg/kg body weight) and 2% Rompun (0.04 mL/kg). The blood was collected from the abdominal artery, and then the quadriceps femoris, hippocampus, and cerebral cortex were extracted. Western blotting was performed as follows to measure the expressions of proteins in each tissue. Total proteins were extracted using RIPA lysis buffer (Cat. #MB-030-0050, Rockland) containing a protease inhibitor cocktail (Cat. #PPI1015, Quartett), and 10 μg of protein was resolved by sodium dodecyl sulfate-polyacrylamide gel electrophoresis and transferred to nitrocellulose membranes using the Trans-Blot Turbo Transfer System (Bio-Rad, Hercules, CA, USA). The membranes were blocked with Tris-buffered saline containing 5% skim milk (Bio-Rad) and 0.2% Tween 20 (Bio-Rad). The following primary antibodies were used: TrkB (1:500, abcam, Cambridge, UK, ab18987), CaMkII (Ca^2+^/calmodulin-dependent protein kinase II) (1:500, Santa Cruz Biotechnology, Santa Cruz, CA, USA, sc5306), AMPK (1:500, abcam, Cambridge, UK, ab133448), PGC-1α (1:500, Cell Signaling Technology, Beverly, MA, USA, 3G6), FNDC5 (1:500, abcam, Cambridge, UK, ab174833), beta-amyloid (1:500, Santa Cruz Biotechnology, Santa Cruz, CA, USA, sc28365), BDNF (1:500, Santa Cruz Biotechnology, Santa Cruz, CA, USA, sc65514), NGF (nerve growth factor) (Biorbyt, Cambridge, UK, DP5931), GDNF (glial cell-derived neurotrophic factor) (1:500, Santa Cruz Biotechnology, Santa Cruz, CA, USA, sc13147) and β-actin (Calbiochem, Darmstadt, Germany). After reaction with horseradish peroxidase-conjugated secondary antibodies (Santa Cruz Biotechnology Inc., Dallas, TX, USA), protein bands were visualized using Clarity Western ECL Substrate (Bio-Rad) following the manufacturer’s procedure. Band densities were determined using a ChemiDoc XRS + System (Bio-Rad) and were normalized to β-actin, which served as a loading control. Plasma glucose (Elabscience, Houston, TX, USA, E-EL-R0430), insulin (Elabscience, Houston, TX, USA, E-EL-R2466), CRP (Elabscience, Houston, TX, USA, E-EL-R0506), MDA (Elabscience, Houston, TX, USA, E-EL-0060), and beta-amyloid (MyBioSource, San Diego, CA, USA, MBS2603593) concentrations were measured using an ELISA kit.2.7. Statistical analysisThe results from each measurement were calculated as mean and standard error (Mean ± SE), and statistical analysis was performed using the SigmaPlot 12.0 statistical package. One-way ANOVA was conducted to verify the differences among the groups of variables, and two-way repeated-measures ANOVA was performed to analyze the group-by-time interactions of OGTT and cognitive function. In this study, after inducing obesity by ingesting a high-fat diet, rats were randomly divided into 4 groups, and 2 groups each were injected with Saline and β-amyloid, and then divided into a group that exercised and a group that did not. A method of comparing the differences between groups was applied by dividing into 4 independent groups according to whether β-amyloid injection and exercise program were performed, one-way ANOVA and a post hoc test for comparison of individual groups were applied for comparison among 4 groups. Tukey’s method was used for the post-hoc test and the statistical significance level was set to α = 0.05.

## 3. Results

### 3.1. Changes in Food Intake and Body Weight

Figure 2 presents the changes in food intake and body weight over the period of obesity induction and the subsequent 12-week intervention. The food intake exhibited similar patterns of change without significant between-group variations during both periods, and in general, the food intake increased in the early phase but slowly decreased after the mid-point of each period. Body weight increased equally across all four groups during the obesity-induction period to reach a significantly higher level by Week 6 compared to Week 1 (*p* < 0.01). Body weight increased in all four groups during the 12-week intervention period, but at Week 12, the S-Ex and β-Ex groups which had followed the exercise program showed significantly lower body weights than the S-Non-Ex and β-Non-Ex groups (*p* < 0.01).

### 3.2. Tissue Weight

A comparison of the tissue weight of the quadriceps femoris and the brain after the 12-week intervention indicated no significant between-group variations, as shown in Figure 3.

### 3.3. Comparison of Cognitive Function and Physical Fitness

Figure 4 presents the change in cognitive function over the 12-week intervention and the comparison of physical fitness after the 12-week intervention. After 12 weeks, short-term memory was significantly improved in the S-Ex and β-Ex groups (*p* < 0.05), but significantly (*p* < 0.05) decreased in the β-Non-Ex group than before the treatment. However, there was no significant difference between the saline group and the β-amyloid group before treatment for 12 weeks.

The level of spatial perception was higher after 12 weeks in the β-Ex group, although no significant difference was observed. Physical fitness, as measured after the 12-week intervention, was significantly higher in the S-Ex and β-Ex groups than in the respective Non-Ex groups (*p* < 0.05).

### 3.4. Changes in Blood Indicators

Figure 5 presents the comparison of the concentrations of blood indicators after the 12-week intervention. All indicators measured (glucose, insulin, CRP, MDA, and β-amyloid) showed significantly lower levels in the S-Ex and β-Ex groups than in the respective Non-Ex groups (*p* < 0.05), while the levels were significantly higher in the β-Non-Ex group than in the S-Non-Ex group (*p* < 0.05). In addition, after the 12-week intervention, blood glucose levels were higher in the Non-Ex groups than in the Ex groups in OGTT, as shown in Figure 6. Notably, blood glucose was significantly higher in the β-Non-Ex group than in the S-Ex group during the 120-min recovery period (*p* < 0.05).

### 3.5. Changes in Protein Expressions of Cytokines in Different Tissues

Figure 7 presents the comparison of protein expression patterns of TrkB, CaMkII, AMPK, PGC-1α, FNDC5, and BDNF in the quadriceps femoris. The levels of TrkB, CaMkII, and AMPK were lower in the β-amyloid groups than in the saline groups, while the Ex groups showed increased expressions of these proteins relative to the Non-Ex groups. The differences, however, were not significant. Likewise, the levels of PGC-1α, FNDC5, and BDNF were lower in the β-amyloid groups than in the saline groups, while the Ex groups showed increased expressions relative to the Non-Ex groups. Notably, the S-Ex group showed significantly higher protein expressions of PGC-1α, FNDC5, and BDNF than the S-Non-Ex group (*p* < 0.05).

Figure 8 presents the comparison of protein expression patterns of TrkB, FNDC5, and BDNF in the hippocampus. The levels of TrkB, FNDC5, and BDNF were lower in the β-amyloid groups than in the saline groups, while the Ex groups showed significantly higher levels than the Non-Ex groups (*p* < 0.05), indicating a clear activation effect of the exercise program.

Figure 9 and Figure 10 present the comparison of protein expression patterns of TrkB, CaMkII, PGC-1α, FNDC5, BDNF, NGF, and GDNF in the cerebral cortex. The levels of most of these markers were lower in the β-amyloid groups than in the saline groups, while the Ex groups showed increased expressions compared to the Non-Ex groups. Notably, the S-Ex group exhibited significantly higher protein expressions of TrkB, CaMkII, PGC-1α, FNDC5, and BDNF than the S-Non-Ex group (*p* < 0.05). The levels of NGF and GDNF were also lower in the β-amyloid groups than in the saline groups, while the Ex groups showed increased expressions compared to the Non-Ex groups, although the differences were not significant. The protein expression of β-amyloid in the cerebral cortex was lower in the Ex groups than in the Non-Ex groups. Notably, the β-Ex group showed a significant decrease in β-amyloid compared to the β-Non-Ex group (*p* < 0.05).

## 4. Discussion

In this study, the change in body weight was a prerequisite to the analysis of how obesity influenced the factors related to brain function. The body weight of the rats in the groups that performed exercise after obesity induction through the 12-week intervention was significantly lower than those in the groups that did not perform exercise, which suggested a positive effect of the exercise program used in this study on the control of body weight. However, the lack of a notable change in body weight until after the middle of the 12-week intervention period indicated that the exercise program used in this study provided a combined stimulation of body fat reduction via aerobic training and muscle mass retention via resistance training [24]. The form of exercise was shown to have almost no effect on the food intake, while the mass of the quadriceps femoris was maintained at a steady level.

In terms of physical fitness after the 12-week intervention, the groups that followed the combined exercise program indicated a positive effect of the exercise program, as reflected in the significantly higher levels of muscular endurance and cardiopulmonary function in these groups. The combined exercise program used in this study was thus verified to have an effect of enhancing both cardiopulmonary and muscle functions [25], thereby fulfilling the prerequisite to analyzing the correlation between improved physical fitness via exercise and the protein expressions of the factors related to brain function.

In this study, the rats injected with β-amyloid exhibited a severe decrease in short-term memory, clearly confirming the role of β-amyloid as a key factor of cognitive dysfunction in aging [26]. Wu et al. [27] guided older adults with cognitive dysfunction to perform exercise, and reduced levels of β-amyloid and Tau were observed, which was taken as evidence to support the effectiveness of exercise in enhancing cognitive function. These results agreed with those of the present study, wherein short-term memory as measured by the passive avoidance test significantly improved in both S-Ex and β-Ex groups that had followed the combined exercise program. After β-amyloid injection and the subsequent combined exercise program, the change in the blood concentration of β-amyloid and the protein expression of β-amyloid in the cerebral cortex were in line with the pattern of change in short-term memory, which clearly confirmed that the exercise program had a positive effect on the control of β-amyloid, a key factor in cognitive dysfunction in aging [26]. Notably, the form of treadmill exercise used in this study has been widely acknowledged as an effective method for the recovery of potential cognitive dysfunction and psychological withdrawal in aging [28,29].

The changes in protein expression patterns of cytokines related to brain function indicated that the combined exercise had clearly activated the protein expressions of PGC-1α, FNDC5, and BDNF in the quadriceps femoris, TrkB, FNDC5, and BDNF in the hippocampus, and PGC-1α, FNDC5, and BDNF in the cerebral cortex. The BDNF expression in the cerebral cortex has been shown to increase after exercise, as the activation of PGC-1α expression during the process of muscle contraction increases the secretion of irisin in the nervous system [30,31]. Hence, the result in this study that demonstrates higher protein expressions of PGC-1α, FNDC5, and BDNF in the quadriceps femoris, hippocampus, and cerebral cortex in the rats after exercise is presumed to verify the positive effect of the combined exercise program on the promotion of brain function. While aerobic exercise for at least two weeks was shown to lead to clear improvements in the factors that promote brain function as well as immunity in older adults [16,32], resistance exercise was reported to have heterogeneous effects across the relative majority of studies [33]. However, the results of the combined program of aerobic and resistance training applied in this study provide evidence to support its alleviating effects on reduced brain function in aging. β-amyloid plaques have been regarded as a critical indicator of early induction of Alzheimer’s disease. In this study, likewise, β-amyloid was clearly shown to be a key factor related to cognitive dysfunction [34], as the protein expressions of PGC-1α, FNDC5, and BDNF in the quadriceps femoris, hippocampus, and cerebral cortex were reduced in the SD rats injected with β-amyloid, which concurred with the decline in short-term memory. These phenomena were also shown to be significantly alleviated through the combined exercise program in this study, which supports the potential use of such an exercise program in dementia prevention [26]. The effect of exercise in reducing β-amyloid plaques was also shown to have a positive role in activating the junction proteins between the hippocampus and cerebral cortex in the nervous system and enhancing cognitive function [35].

AMP-activated protein kinase (AMPK) has a role in maintaining energy homeostasis in the body through exercise stimulation, and is regarded as a critical factor in cell viability and neurogenesis [36]. CaMkII, as a regulator of mitochondrial biogenesis for the activation of energy production in the body, has a role in controlling the PGC-1α deacetylation mechanism [37]. Both CaMkII and AMPK are activated during exercise, and they act as essential upstream factors that activate the transcription factors PGC-1α and FNDC5 [38]. When the levels of CaMkII and AMPK are reduced at the onset of neurodegenerative disease, the consequent reduced ability to express PGC-1α, FNDC5, BDNF, NGF, and GDNF could lead to cognitive dysfunction and deteriorating physical fitness [39]. In addition, while SD rats injected with β-amyloid showed reduced protein expressions of NGF and GDNF in the hippocampus and cerebral cortex [40], the exercise program was shown to restore the protein expression levels of NGF and GDNF [41]. In this study, the expression levels of proteins such as CaMkII, AMPK, NGF, and GDNF in the tissues of rats treated with β-amyloid injection and exercise program showed a similar trend to the results of previous studies, but no significant difference was observed. However, reductions in the levels of all factors in each tissue of the SD rats after the injection of β-amyloid and the restoration of the levels following the combined exercise program were both consistent, which suggests a positive effect of the combined exercise program in terms of promoting brain function.

## 5. Conclusions

In this study, obese SD rats injected with β-amyloid showed consistent improvements in cardiopulmonary function, muscular endurance, and short-term memory after following a 12-week exercise program combining aerobic and resistance training. The positive effects of the combined exercise program were attributed to the alleviation of β-amyloid plaques, a key factor involved in the reduced brain function, and to the activation of protein expressions of PGC-1α, FNDC5, and BDNF in the quadriceps femoris, hippocampus, and cerebral cortex.

## Figures and Tables

**Figure 1 ijerph-19-07921-f001:**
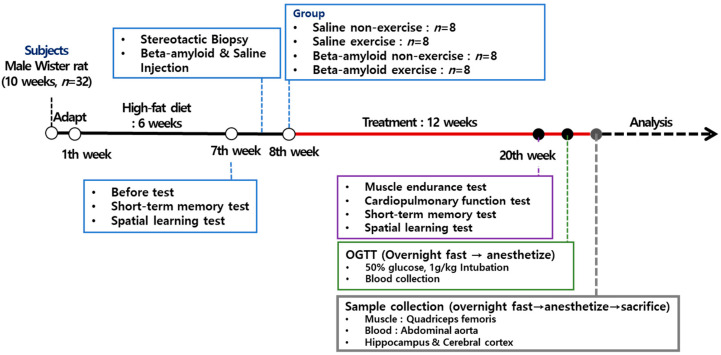
Schematic diagram of experimental procedures.

**Figure 2 ijerph-19-07921-f002:**
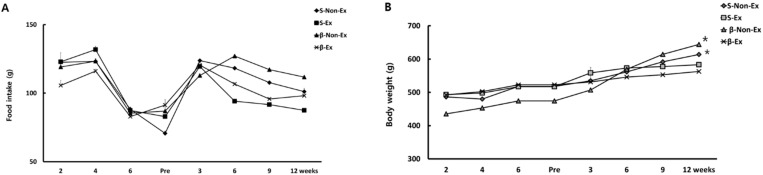
Changes in food intake (**A**) and body weight (**B**) during the obesity-induction period and the 12-week intervention period. (* *p* < 0.05 Significant difference as compared to Non-Ex group).

**Figure 3 ijerph-19-07921-f003:**
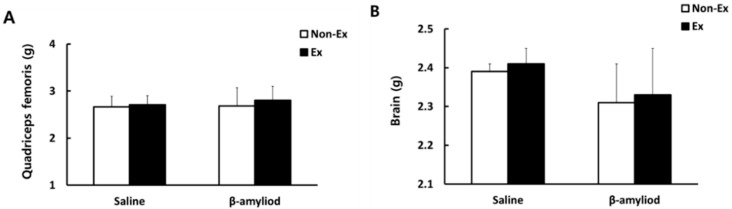
Comparisons of the weight of the quadriceps femoris (**A**) and brain (**B**) after the 12-week intervention.

**Figure 4 ijerph-19-07921-f004:**
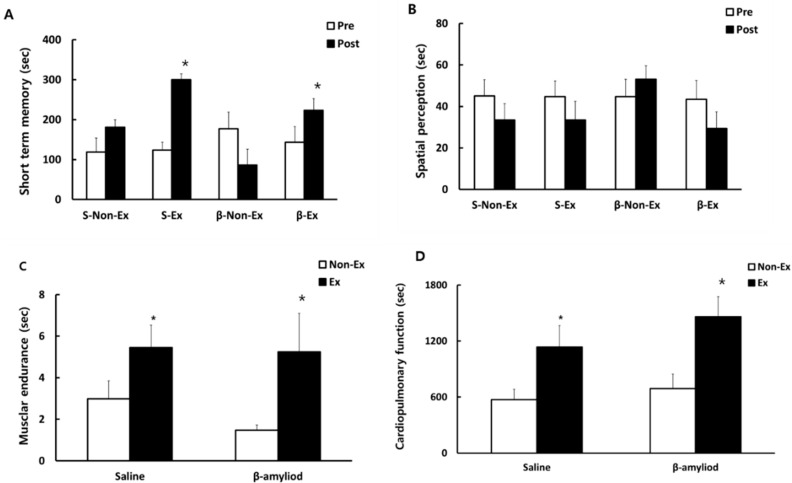
Scores of short-term memory (**A**) and spatial perception (**B**) before and after the 12-week intervention, and the comparison of muscular endurance (**C**) and cardiopulmonary function (**D**) after the 12-week intervention. (* *p* < 0.05 Significant difference as compared to Non-Ex group).

**Figure 5 ijerph-19-07921-f005:**
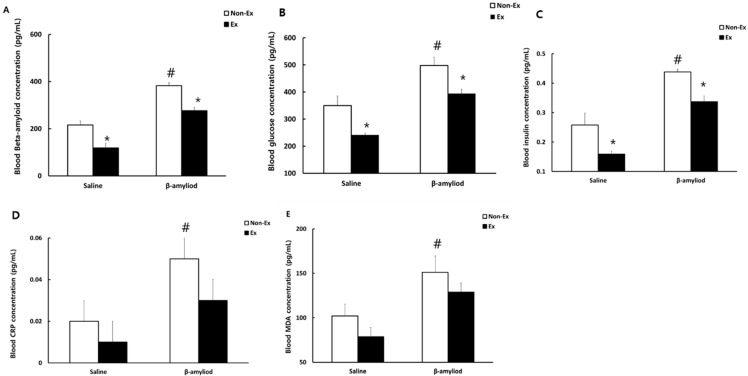
Comparisons of blood concentrations of (**A**) β-amyloid, (**B**) glucose, (**C**) insulin, (**D**) CRP, and (**E**) MDA after the 12-week intervention. (* *p* < 0.05 Significant difference as compared to Non-Ex group; # *p* < 0.05 Significant difference as compared to Saline group).

**Figure 6 ijerph-19-07921-f006:**
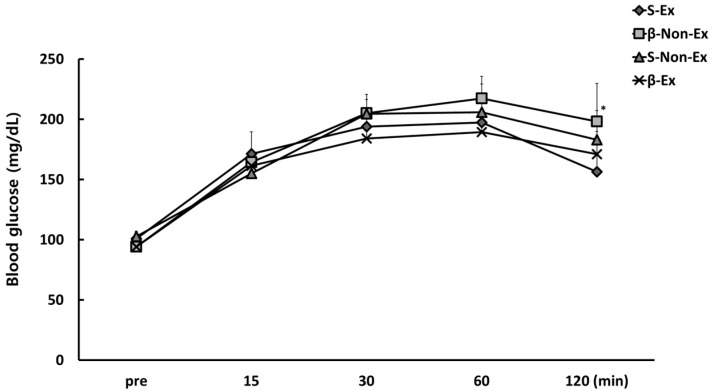
Comparison of blood concentrations of glucose in OGTT after the 12-week intervention. (* *p* < 0.05 Significant difference as compared to Ex group).

**Figure 7 ijerph-19-07921-f007:**
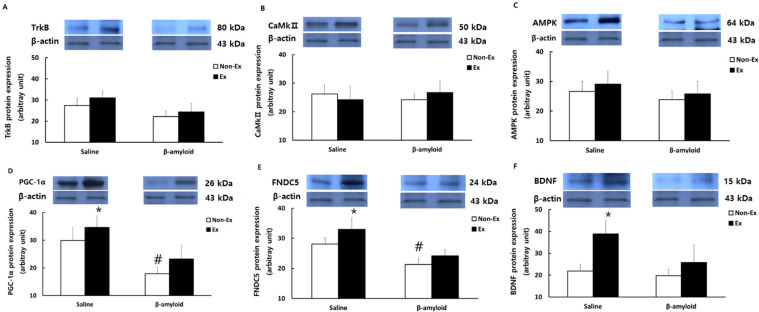
Comparisons of protein expressions of (**A**) TrkB, (**B**) CaMkII, (**C**) AMPK, (**D**) PGC-1α, (**E**) FNDC5, and (**F**) BDNF in the quadriceps femoris after the 12-week intervention. (* *p* < 0.05 Significant difference as compared to Non-Ex group; # *p* < 0.05 Significant difference as compared to Saline group).

**Figure 8 ijerph-19-07921-f008:**
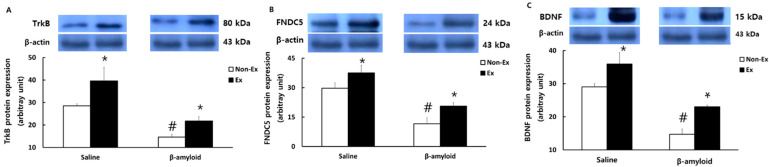
Comparisons of protein expressions of (**A**) TrkB, (**B**) FNDC5, and (**C**) BDNF in the hippocampus after the 12-week intervention. (* *p* < 0.05 Significant difference as compared to Non-Ex group; # *p* < 0.05 Significant difference as compared to Saline group).

**Figure 9 ijerph-19-07921-f009:**
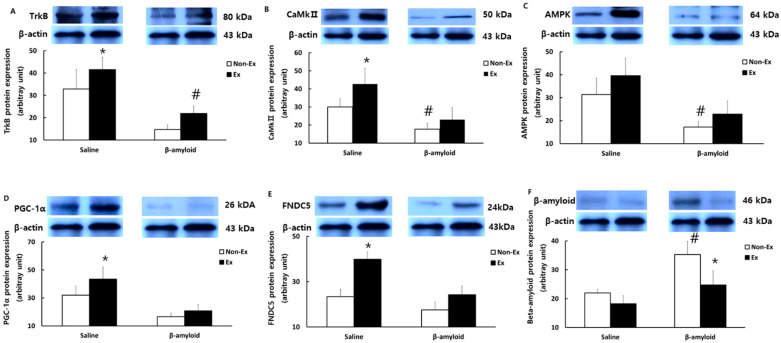
Comparisons of protein expressions of (**A**) TrkB, (**B**) CaMkII, (**C**) AMPK, (**D**) PGC-1α, (**E**) FNDC5, and (**F**) β-amyloid in the cerebral cortex after the 12-week intervention. (* *p* < 0.05 Significant difference as compared to Non-Ex group; # *p* < 0.05 Significant difference as compared to Saline group).

**Figure 10 ijerph-19-07921-f010:**
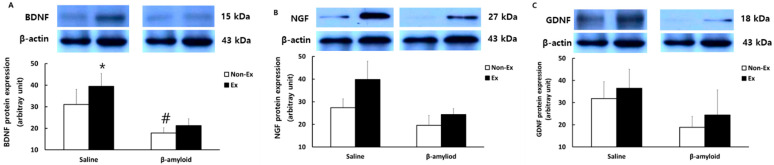
Comparisons of protein expressions of (**A**) BDNF, (**B**) NGF, and (**C**) GDNF in the cerebral cortex after the 12-week intervention. (* *p* < 0.05 Significant difference as compared to Non-Ex group; # *p* < 0.05 Significant difference as compared to Saline group).

## Data Availability

Not applicable.

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
