# Peer review of "Effects of an Exercise Program Combining Aerobic and Resistance Training on Protein Expressions of Neurotrophic Factors in Obese Rats Injected with Beta-Amyloid"

_ijerph, 2022, doi:10.3390/ijerph19137921_

Round 1
Reviewer 1 Report
The manuscript is well written and presented.
However, I suggest to the author complete the sentences line 43:
neurodegenerative diseases are also correlated with low androgen and estrogen levels (Bianchi VE et al. J Endocr Soc. 2020; Vegeto, E. et al Endocr Rev. 2020 ).
Furthermore, it should be explained why the authors choused the exercise protocol aerobic+resistance exercise, and not aerobic versus resistance exercise.
Author Response
Response to Reviewer 1 Comments
The manuscript is well written and presented.
Point-1: However, I suggest to the author complete the sentences line 43:
neurodegenerative diseases are also correlated with low androgen and estrogen levels (Bianchi VE et al. J Endocr Soc. 2020; Vegeto, E. et al Endocr Rev. 2020 ).
Response-1: In accordance with the points pointed out, the following modifications have been made and references have been added.
Neurodegenerative diseases associated with beta-amyloid plaques show strong correlations with obesity, higher insulin resistance, and metabolic diseases [3].
- Neurodegenerative diseases associated with beta-amyloid plaques show strong correlations with obesity, higher insulin resistance, metabolic diseases [3], low androgen and estrogen levels [4,5].
- Bianchi,V. E.; Rizzi,L.; Bresciani, E.; Omeljaniuk, R. J.; Torsello, A. Androgen therapy in neurodegenerative diseases. Journal of the Endocrine Society 2020, 4(11), bvaa120, https://doi.org/10.1210/jendso/bvaa120
- Vegeto, E.; Villa, A.; Torre, S. D.; Crippa, V.; Rusmini, P.; Cristofani, R.; Galbiati, M.; Maggi, A.; Poletti, A. The role of sex and sex hormones in neurodegenerative diseases. Endocr Rev.2020, 41(2), 273-319.
Point-2: Furthermore, it should be explained why the authors choused the exercise protocol aerobic+resistance exercise, and not aerobic versus resistance exercise.
Response-2: In consideration of the points pointed out, the following sentences have been added and references have been added.
It is noteworthy that the combination of aerobic and resistance exercise has been shown to contribute to the prevention of sarcopenia, a key factor in the maintenance of health as one ages, and in the activation of cardiopulmonary function, while exerting parallel effects on the activation of the metabolic functions of factors related to brain function and on the secretion of myokines. In addition, it is required to combine high-intensity aerobic exercise and resistance exercise when performing exercise to prevent brain function deterioration in the aging process [16,17].
- De la Rosa, A.; Olaso-Gonzalez, G.; Arc-Chagnaud, C.; Millan, F.; Salvador-Pascual, A.; García-Lucerga, C.; Blasco-Lafarga, C.; Garcia-Dominguez, E.; Carretero, A.; Correas, A.G.; Vina, J.; Gomez-Cabrera,C. Physical exercise in the prevention and treatment of Alzheimer’s disease.J Spt Health Science 2020, 9, 394-404.
- Northey, J.M.; Cherbuin, N.; Pumpa, K.L.; Smee, D.J.; Rattray, B. Exercise interventions for cognitive function in adults older than 50: a systematic review with meta-analysis. Br J Sport Med 2018, 52, 154-160.

Reviewer 2 Report
Alzheimer’s disease (AD) has been reported as a multi-factorial disorder and obesity with an increased level of beta-amyloid plaques in brain structures is associated with the pathogenesis of the disease. The authors riced the right question of the physical exercises could be preventing factor in the occurrence of AD or if the combined exercises could be effective in the treatment of rats in an experimental AD-like model.
Nevertheless, the analysis of the obtained results is not adequate for the research model used. The authors used one-way ANOVA instead of Factorial ANOVA (in the experiments 2 factors were involved: exercises and beta-amyloid injections), so the statistical significance can be different and the conclusions as well. Moreover, all factors tested are compared to the control group (saline, no exercise group). Why did the authors not show the effect of exercise in the model (β-Non-Ex group vs. β-Ex group), and above all, the effect of the model itself (no comparisons of the group receiving saline to the group receiving beta-amyloid)?
The authors should also have included a diagram of the research procedures used over time as it is not clearly described.
The results presented in this way do not answer the question of whether the AD model is well-chosen and how exercise influences the course of the disease.
Author Response
Response to Reviewer 2 Comments
Alzheimer’s disease (AD) has been reported as a multi-factorial disorder and obesity with an increased level of beta-amyloid plaques in brain structures is associated with the pathogenesis of the disease. The authors riced the right question of the physical exercises could be preventing factor in the occurrence of AD or if the combined exercises could be effective in the treatment of rats in an experimental AD-like model.
Point-1: Nevertheless, the analysis of the obtained results is not adequate for the research model used. The authors used one-way ANOVA instead of Factorial ANOVA (in the experiments 2 factors were involved: exercises and beta-amyloid injections), so the statistical significance can be different and the conclusions as well. Moreover, all factors tested are compared to the control group (saline, no exercise group). Why did the authors not show the effect of exercise in the model (β-Non-Ex group vs. β-Ex group), and above all, the effect of the model itself (no comparisons of the group receiving saline to the group receiving beta-amyloid)?
Response-1: We think that the reviewer’s comments are a part that can be sufficiently suggested. It is acknowledged that the experimental design of this study was a design that did not fully consider the points pointed out by the reviewer.
However, while explaining the experimental plan of this study, the following contents were added in detail in the statistical methods and result explanations from the point of view of related supplementation. These revisions explain the results of statistical processing in more detail and are intended to supplement the points pointed out by the reviewer.
Response-1-1: Added contents to statistics analysis: In this study, after inducing obesity by ingesting a high-fat diet, rats were randomly divided into 4 groups, and 2 groups each were injected with Saline and β-amyloid, and then divided into a group that exercised and a group that did not. A method of comparing the differences between groups was applied by dividing into 4 independent groups according to whether β-amyloid injection and exercise program were performed, one-way ANOVA and a post hoc test for comparison of individual groups were applied for comparison among 4 groups.
Response-1-2: Significant differences were indicated in the graph according to the sentences explained in the weight comparison at week 12.
Response-1-3: Some of the result descriptions have been modified as follows.
After 12 weeks, short-term memory was significantly improved in the S-Ex and β-Ex groups (p<0.05), but significantly worse in the β-Non-Ex group (p<0.05).
- After 12 weeks, short-term memory was significantly improved in the S-Ex and β-Ex groups (p<0.05), but significantly (p<0.05) decreased in the β-Non-Ex group than before the treatment. However, there was no significant difference between the saline group and the β-amyloid group before treatment for 12 weeks.
The levels of TrkB, FNDC5 and BDNF were lower in the β-amyloid groups than in the saline groups, while the Ex groups showed significantly higher levels than the Non-Ex groups (p<0.05).
- The levels of TrkB, FNDC5 and BDNF were lower in the β-amyloid groups than in the saline groups, while the Ex groups showed significantly higher levels than the Non-Ex groups (p<0.05), indicating a clear activation effect by the exercise program.
Point-2: The authors should also have included a diagram of the research procedures used over time as it is not clearly described.
Response-2: A diagram of the overall experimental procedures has been added as follows.
Figure 1. Schematic diagram of experimental procedures

Reviewer 3 Report
This article provides evidence that exercise is effective in obese SD rats injected with β-amyloid.
Comments and suggestions:
1. Most bands are very faint in Figure 7, which make the quantification difficult and not convincing.
2. Can the authors show immunofluorescence images of brain that how exercise change the pattern of β-amyloid expression in β-Ex group versus β-Non-Ex group?
Author Response
Response to Reviewer 3 Comments
This article provides evidence that exercise is effective in obese SD rats injected with β-amyloid.
Comments and suggestions:
Point-1: Most bands are very faint in Figure 7, which make the quantification difficult and not convincing.
Response-1: In consideration of the points you pointed out, Figure 7 has been modified and supplemented.
Figure 8(revised from Figure 7). Comparisons of protein expressions of (A) TrkB, (B) FNDC5, and (C) BDNF in the hippocampus after the 12-week intervention. (*p<0.05 Significant difference as compared to Non-Ex group; #p<0.05 Significant difference as compared to Saline group)
Point-2: Can the authors show immunofluorescence images of brain that how exercise change the pattern of β-amyloid expression in β-Ex group versus β-Non-Ex group?
Response-2: Unfortunately, immunofluorescence images of the brain were not attempted in this study. In this study, we suggested change the pattern of β-amyloid expression by comparing the β-amyloid concentration and β-amyloid protein expression in blood between the β-Ex group versus the β-Non-Ex group. In the next study, we will actively try the immunofluorescence images analysis of the brain. Thank you so much.

Round 2
Reviewer 2 Report
Although, the Authors improved some points and described the research procedures, in my opinion using one-way ANOVA and post-hoc (for comparison between two groups) is pointless. According to my best knowledge, when the experiment is conducted with two different factors (beta-amyloid and exercises) the factorial ANOVA should be used (like in the ref. no 6, you cited). Otherwise, the results obtained in exercised groups are not comparable between the two used models. Moreover, when the Authors cite the data about protein level (i.e. lines 66, 67) they should rather cite the original instead of the review article.
In my opinion, the statistical analysis should be recalculated. Therefore, I leave the decision to accept the work for publication to the editor of the journal.